



# Dynamics of snow melt infiltration into mountain soils: an instrumental approach in the Nant Valley, Swiss Alps

Judith Eeckman[1], Brian De Grenus[2], Floreana Miesen[2], James Thornton[4], Philip Brunner[5], and Nadav Peleg[2,3]

[1]Institute of Geography and Sustainability, University of Lausanne, Lausanne, Switzerland
[2]Institute of Earth Surface Dynamics, University of Lausanne, Lausanne, Switzerland
[3]Expertise Center for Climate Extremes, University of Lausanne, Lausanne, Switzerland
[4]Mountain Research Initiative, c/o University of Bern, Switzerland
[5]Centre d'Hydrogéologie et de Géothermie (CHYN), Université de Neuchâtel, Switzerland

**Correspondence:** Judith Eeckman ju.eeckman@gmail.com, judith.eeckman@unil.ch

**Abstract.** To gain a deeper understanding of the dynamics of the contribution of snow melt to mountainous water cycles, it is necessary to better grasp the parameters controlling the infiltration of snow melt into mountainous soils. This research uniquely combines snow melt rate data with soil moisture dynamics, providing a comprehensive, three-year dataset. The integration of multiple measurement techniques and the estimation of the snow melt rate through the measurement of snow resistivity offer
5  a new perspective on snow melt infiltration processes. The study area is located in the Nant Valley, Swiss Alps. Measurement points are distributed in mid to high elevations in various alpine environments. Besides demonstrating the instrumental setup, we also investigated the snowmelt-infiltration dynamics in the study area. Results indicate that, even though melt rates are considerably lower than soil saturated hydraulic conductivity values (with a ratio of $3.1 \times 10^{-3}$ on average), the response times of shallow soil moisture and stream discharge to melt events is fast (from 2 to 5 hours). At the point measurement, snowmelt
10  hardly infiltrates below 30 cm. These findings emphasize the potential vulnerability of mountain areas to dry periods in the future, particularly in the context of the expected shortening of the melt period due to climate change.





## 1 Introduction

Since snowmelt infiltration toward the subsurface can contribute significantly to soil water content, snowmelt plays an important role in mitigating water stress during dry periods in mountainous regions (Niu and Yang, 2006; Bayard et al., 2005; Zappa and Kan, 2007; Brunner et al., 2021). In the context of global climate change, the amount of solid precipitation is projected to decrease and the onset of the melt is expected to commence earlier (Masson-Delmotte et al., 2021), changing the snowmelt-infiltration dynamics. In the Swiss Alps, for example, a reduction of snow amount, a shortening of the melt period, and faster melt rates associated with shortened snow events are expected (Fischer et al., 2022). Therefore, it is essential to gain a better understanding of the parameters that control snowmelt infiltration into the soil.

Knowledge of the preferential pathways of snowmelt flux in the surface and subsurface remains illusive (Fang et al., 2019). Due to steep slopes and erosion processes, alpine soils are generally relatively thin (with depths generally smaller than 1 m) and commonly present textures from sand to silt (Legros, 1992). Infiltration processes in mountainous soils are controlled by two competing factors: (i) the coarse granulometry of superficial soils, which enhances their infiltration capacity (Legros, 1992); and (ii) the typically steep slopes, which increase the velocity of lateral transfer (Webb et al., 2018b; Carey and Woo, 2001). Kampf et al. (2015) provide an overview of the different snowmelt infiltration processes encountered in various mountainous areas. Evidence of fast lateral transfer in shallow soils during snowmelt periods is mentioned in several works (Santos et al., 2018; Fang et al., 2019; Heidbüchel et al., 2012). Young waters (i.e., from snowmelt and superficial storage) have been shown to actively contribute to discharge during both winter and spring melt periods (Ceperley et al., 2020). An "inverse storage effect" (i.e., emptying of the most superficial soil layers) has been observed during snow accumulation periods (Benettin et al., 2017; Wilusz et al., 2020). However, in many mountain settings, groundwater and saturated zone processes are also important. Circulations in deeper layers, particularly through unconsolidated moraine deposits and fractured bedrock, are observed in mountainous catchments (Schaefli et al., 2014; Meeks et al., 2017; Thornton et al., 2018).

This work aims to detail the physical processes involving the infiltration of snow melt at a study point through the following scientific questions: (i) Comparing the intensities of melt rates with the infiltration capacity of soils to better quantify the partition between surface runoff and infiltration of the melt flux; (ii) Understanding the vertical percolation of the melt flux into soil layers to better describe the response of the soil column to melt events; and (iii) Describing the dynamics of lateral transfers of the flux issued from snow melt in the superficial soil layers and the response of the stream discharge to melt events to better quantify the response of the catchment to melt events.

Despite a wide range of model formalisms, from simple temperature index methods or fuller energy balance methods, being applied to simulate the flux at the interface between snowpack and soil (Martinec, 1975; Rulin et al., 2008; Vionnet et al., 2011; He et al., 2014; Zhang et al., 2015), modeling approaches keep facing large uncertainties in representing accurate snowmelt infiltration into soils. This is mainly due to large heterogeneity in soil and snowpack properties and lack of measurements for these particular variables in mountainous areas (Meeks et al., 2017). Some studies apply direct monitoring of meltwater using snow lysimeters, from simple buried rain gauges to complex melted water collecting systems (Kattelmann, 2000; Webb et al., 2018a). This method presents the advantage of directly measuring the variable of interest, although the instrument either




remains vulnerable to an excess load of snow or requires an extended resistant structure which often cannot be installed in remote mountain areas. Other studies estimate the melt rate through the variation of snow depth or snow water equivalent of the snowpack (Kampf et al., 2015; Fang et al., 2019; Archer and Stewart, 1995). However, these variables take into account neither the vertical heterogeneity of the snowpack nor surface effects such as snow sublimation or wind depletion. Here, we

propose monitoring the snow melt rate based on a direct measurement of the liquid water content of the snowpack through snow electric resistivity. This instrumental method has previously been shown to yield reliable assessments of snow melt rates in various alpine studies(French and Binley, 2004; Gance et al., 2016; Bloem et al., 2020).

The vertical infiltration of snow melt into soil layers and its lateral transmission along the slopes, depending upon soil properties and structure, is another aspect that is difficult to estimate. One potential assessment method is using natural tracer

analysis, in particular water stable isotopes (Klaus and McDonnell, 2013; Beria et al., 2018; Michelon et al., 2022). However, the transfers into superficial soils are hard to detect via natural tracers because their signature is inconclusive. We suggest monitoring snowmelt infiltration into soil layers at depths up to 30 cm using a network of capacitive probes. These probes can be deployed at the same locations where snowmelt is monitored, allowing for the tracking of the vertical penetration of flux melt into soil layers and accounting for spatial variations.

The proposed novel monitoring setups can provide valuable insights into snowmelt-infiltration dynamics in mountainous catchments. This approach will enhance our understanding and help assess the role of snowmelt and soil water storage in mitigating downstream effects of dry periods, offering novel and generalizable data. We present an implementation of our monitoring approach in the Nant Valley. The monitoring is performed for three consecutive years (2021 to 2024) and winter field campaigns are conducted to provide validation measurement of snow properties.

## 65 2 Study area

### 2.1 Site description

The study area is located in the Nant Valley (Vallon de Nant; 46.23°N, 7.07°E), a Swiss pre-Alpine catchment that contains typical alpine ecosystems, from deciduous forest to post-glacial recolonization (Fig. 1). The catchment is chosen because of its importance for ecological monitoring and the quantity and duration of measurements available from decades of monitoring.

Perret and Martin (2015) presented a detailed map of surface geomorphological units. Three main geomorphological units are described (Fig. 1B): (i) limestone cliffs from the Nappe de Morcles on the east ridge of the valley; (ii) active and passive moraine deposits from the Martinet glacial on the upper part of the valley; and (iii) flysch cliffs and associated screes and rockfalls on the west ridge of the valley. The majority of the superficial soils in the Nant Valley are developed on top of ancient moraine deposits, screens, and landslides, resulting in sandy to silty, relatively shallow soils (Grand et al., 2016). These formations are

common in alpine areas, however, the accumulation of erosion material from the schist cliffs into small depressions led to the development of relatively locally deeper soils in the catchment. Several studies investigated the catchment's surface and subsurface hydrology (Antoniazza, 2023; Michelon et al., 2023; Thornton et al., 2022). Besides assessing the recent shift in





**Figure 1.** The Nant valley, Bex, Switzerland, and the locations of the instrument set up deployed in this study (A), together with (B) geomorphological units described by (Perret and Martin, 2015) and (C) Corine Land Cover 2006 (Aune-Lundberg and Strand, 2010). The point numbers correspond to sample points in Table 1.

snowmelt peak and its impact on the discharge, these studies showed that a limited understanding of snowmelt flow paths hinders hydrological model development for this catchment, which further motivated our research in this area.

## 2.2 Meteorological and hydrological data

A hydrometric station is located at the outlet of the catchment (Fig. 1A), recording hourly water levels since 2010. A rating curve has been computed based on 55 reference gauging performed at various water levels (Antoniazza, 2023). Moreover, weather data from three weather stations at different locations (at the elevations 1253 m a.s.l., 1485 m a.s.l. and 1780 m a.s.l., respectively) in the catchment is available since 2010 (Fig. 1A). These stations record total precipitation, near-surface

air temperature, atmospheric pressure, shortwave solar radiation, and wind direction and velocity at 5 minutes intervals, which





have subsequently been averaged to hourly timesteps. However, considering the exposure of these stations to harsh climatic conditions and their difficult access, the reliability of the recorded data is low and the time series presents many gaps. Moreover, as the climate stations are not heated, solid precipitation records are considerably underestimated (Benoit et al., 2018; Thornton et al., 2021). Consequently, the estimation of precipitation in the catchment remains highly uncertain. The daily remote sensed

information on Snow Cover Area (SCA) is available at the 500 m resolution from the MODISA1 Level3 remote product (Hall et al., 2009). This product have been used in this work to describe the overall dynamics of the snow cover at the catchment scale, however, its spatial and temporal resolutions are not sufficient enough to compare this product at the pixel scale to the in-situ snow measurement.

## 3 Monitoring and data collecting methods

Three different types of monitoring devices were deployed to obtain local values of snow depth, extent of snow cover, snow water equivalent ($S_{WE}$), and snowmelt rate.

### 3.1 Snow Melt Analyzer

The Snow Melt Analyzer (SMA) is a unique system for automatic and continuous measurements of diverse snowpack parameters developed by the SOMMER Messtechnik company. This monitoring device has been used in various studies in alpine

environments (Sommertechnik, 2009). It can be used by itself or in combination with a more complete instrumental setup for snowpack analysis. Here, the configuration of the instrument that allows the measurement of the liquid and solid water content and density of the bottom snow layer is chosen. We assume the liquid water content of the bottom snow layer ($L_{WC}^{bottom}$, lowermost 7 cm) as a proxy for the snowmelt rate. The SMA instrument consists of a metal frame set up in a suitable position on level ground (Fig. 1D). A weather and UV-resistant sensor band penetrates the snow and measures the volumes of ice, water,

and air content in the snowpack using the variation of impedance between two connected electrodes.

Three aspects were considered in choosing the device location: (i) installing the SMA on flat ground with snow conditions representative of the area and orienting the frame to avoid wind effects; (ii) selecting soils with significant storage and infiltration capacity, specifically deep alpine meadow soils; and (iii) ensuring ease of access and power supply. Consequently, the SMA was deployed near the Auberge climate station at 1253 m a.s.l. This mid-altitude location, with its developed soil, allows

for the analysis of snowmelt rate and represents a favorable case for snowmelt infiltration compared to typical alpine sites with finer soils and steeper slopes.

### 3.2 SnowTree and visual scale

To estimate the snow cover extent in the surroundings of the measurement points, three visual scales coupled with time-lapse cameras were installed in the vicinity of the three climate stations (Fig. 1A). The images obtained from the cameras

were evaluated in two respects: (i) the graduation reached on the visual scale, which gave the local snow depth with a 10 cm accuracy; and (ii) the qualitative extent of snow cover over the visible landscape, divided into three categories: *no snow*,





*partially covered surface*, and *covered surface*. Despite the limited nature of such data, they provide valuable information about snow conditions during the measurement periods.

In addition, to locally assess the snow depth, two "SnowTree" instruments were developed and deployed at the Auberge and Chalet measurement points (Fig. 1A). The SnowTree is a 2.5 m high wooden mast, equipped with small *iButton* thermometers glued every 5 or 10 cm. This small sensor presents good performance for environment science application (Hubbart et al., 2005). Temperatures were recorded every two hours. This instrument aims to track the snow depth by discriminating between thermometers covered by snow or not covered, with a $\pm 5$ cm accuracy. This instrument complements the observations made with the visual scale, which a a vertical accuracy of $\pm$ 20 cm. In addition, this instrument is simpler to install than an optical snow depth sensor, because it does not require any structure or power supply. Reusser and Zehe (2011) propose using the standard deviation of the hourly temperatures computed over 24 hours ($24\ h\ STD$) for this differentiation, as the diurnal amplitude of temperatures is lowered when a sensor is covered by snow. Empirical thresholds were fixed following snow depth values observed at the visual scales:

$$\begin{cases} \text{snow}, & \text{if } T_{day} < 1^\circ C \text{ and } 24\ h\ STD < 4.5 \\ \text{snow}, & \text{if } T_{day} > 1^\circ C \text{ and } 24\ h\ STD < 1.4 \\ \text{no snow}, & \text{otherwise}, \end{cases} \tag{1}$$

where $T_{day}$ [°C] is the average daily temperature and $24\ h\ STD$ [°C] is the standard deviation of the hourly temperatures computed for each day. The two SnowTrees and the visual scales were installed between November 2022 and March 2023 (2022-2023 winter) but were not maintained for the 2023–2024 winter (lack of manpower). Instead, a classical optical infrared snow depth sensor was installed at the Auberge meteorological station in November 2023, which recorded hourly.

### 3.3 Cosmic Ray Sensor

Hydroinnova's Cosmic Ray Sensor (CRS) monitoring device (Fig. 1) was installed at the Auberge station to measure snow water equivalent ($S_{WE}$) over a uniquely large footprint. The advantages of the device, beyond the large footprint, are that it is automatic, easy to install, and requires little maintenance. The basis of the technique is that hydrogen contained in the snowpack attenuate downward neutrons coming from cosmic rays. The amount of attenuation is directly related to the mass of intervening snow, and by extension the amount of $S_{WE}$. The method described by Desilets (2017) to convert incoming neutrons count into $S_{WE}$ value was used:

$$S_{WE} = -\Lambda \ln \frac{N}{N_0}, \tag{2}$$

and

$$N = f_{sol} N_{raw} \exp\left[(P - P_0)\beta\right], \tag{3}$$

where $N_{raw}$ [count h$^{-1}$] is the measured incoming neutrons count, $P$ [hPa] is the atmospheric pressure, $P_0$ [hPa] is the reference barometric pressure and $N_0$ [count h$^{-1}$] is the proton flux in the absence of snow. $\Lambda$ [-] and $\beta$ [-] are fixed parameters,





following the technical recommendations (KIT, 2015). To complement these $S_{WE}$ time series, local density measurements were carried out during the winter of 2022-2023 near the Auberge station, using the method of weighing cylindrical samples: snow profiles were dug, and snow samples were collected snow profiles were dug, and snow samples were collected horizontally with a metal cylinder of volume 550 cm$^3$. The collected snow is weighed to calculate the density of the snow sample. At

each location, a sample is collected approximately 7 cm from the ground (which corresponds to the SMA measurement height ) and, if the depth of the snowpack allows, another sample is taken in the middle of vertical.

### 3.4 Soil sampling and analysis

To describe the variety of soils in the study area, eight sampling sites were chosen. The physiographic characteristics of this eight sites and the instrumental set up are presented on Table 1 and the point ID are referenced on Fig. 1). At each site, soil

cores were taken and the granulometry of the sampled soil were analysed. Infiltrometery tests have been conducted for seven of the eight sampling sites (expect the Bastion point for technical reasons at this distant point). Soil moisture probes have been installed at three of this eight sampling sites that are located close from the climatic stations : the Auberge, Petit Pont and La Chaux points (see Table 1). The locations are chosen to represent the different geomorphological characteristics and environments in the catchment, from deep soil covered by mixed forest to shallow soils developed above the moraine deposit.

In particular, four points at different altitudes (Combe, La Chaux, Petit Pont, and Argile) were specially analysed because they present particularly deep soils, resulting from the accumulation of material eroded from the schist cliffs on the western side of the valley. Evidence of superficial water saturation in spring observed at these sites for the three years studied motivates the detailed analysis of the dynamics of their contribution to the hydrological system. Both grazed and non-grazed sites were sampled. 5TM/DECAGON capacitive sensors were used to measure soil moisture and soil temperature at hourly intervals at

different depths at three points in the catchment (Fig. 1). These sensors were installed in August 2021.

### 3.5 Granulometry analysis and pedotransfer functions

Vertical description of the soils and sampling was performed at one or two auger holes for each sampling point. Granulometry analysis was then performed for each of the samples collected for the main observed horizons at the nine sampling points. Particle size distributions were determined via laser granulometry analysis (Blott et al., 2004). The empirical pedotransfer

functions proposed by Clapp and Hornberger (1978) were used to compute values of soil water content at saturation ($w_{sat}$ [m$^3$ m$^{-3}$]), wilting point ($w_{wilt}$ [m$^3$ m$^{-3}$]), and field capacity ($w_{fc}$ [m$^3$ m$^{-3}$]) from the clay $C$ [-] and sand $S$ [-] fraction of each sample:

$$w_{sat} = (-1.08S + 494.305) \cdot 10^{-3}, \tag{4}$$

$$w_{wilt} = 37.1342 \cdot 10^{-3}C^{0.5}, \tag{5}$$



**Table 1.** Physiographic characteristics of the nine soil sampling points: location, soil depth, geomorphology (Perret and Martin, 2015) and CLC2006 Land Cover (Aune-Lundberg and Strand, 2010) classifications, together with the estimation of soil parameters: granulometry, water content of the total soil column at saturation $w_{sat}$ and at the wilting point $w_{wilt}$, field capacity $w_{fc}$ and hydraulic conductivity at saturation $K_{sat}$. AD - Alluvial deposit; GT - Glacial till; SR - Screes and rockfall; CF - Coniferous forest; NG - Natural grasslands; BR - Bare rock.

| | Auberge | Pissenlit | Chalet | Petit Pont | Protegée | LaChaux | Combe | Bastion |
|---|---|---|---|---|---|---|---|---|
| Point ID | 1 | 2 | 3 | 4 | 5 | 6 | 7 | 8 |
| **Physiographic characteristics** | | | | | | | | |
| Lat/Lon [°N/°E] | 46.251/7.110 | 46.247/7.106 | 46.229/7.102 | 46.231/7.102 | 46.230/7.101 | 46.229/7.092 | 46.225/7.088 | 46.218/7.077 |
| Elevation [m.a.s.l.] | 1257 | 1281 | 1491 | 1473 | 1479 | 1777 | 1853 | 2497 |
| Soil depth [cm] | 69 | 40 | 32 | >130 | 10 | >130 | 90 | 65 |
| Geomorphology | AD | AD | AD | AD | AD | GT | GT | SR |
| CLC2006 | CF | CF | NG | NG | NG | NG | NG | BR |
| Pasture | No | Yes | Yes | Yes | No | Yes | No | No |
| **Instrumental devices** | | | | | | | | |
| Climatic station | ✓ | - | ✓ | - | - | ✓ | - | - |
| SnowTree and visual scale | ✓ | - | ✓ | - | - | - | - | - |
| 5TM sensor depth [cm] | 5, 10, 20, 30 | - | - | 25 | - | 25 | - | - |
| Infiltrometry tested | ✓ | ✓ | ✓ | ✓ | ✓ | ✓ | ✓ | - |
| **Estimation of soil parameters** | | | | | | | | |
| Clay [%] | 14.9 | 17.3 | 14.1 | 15.5 | 9.1 | 13.1 | 15.15 | 16.2 |
| Silt [%] | 60.5 | 70.9 | 70.2 | 67.3 | 51.9 | 62.7 | 62.1 | 63.2 |
| Sand [%] | 24.6 | 11.7 | 15.7 | 17.2 | 38.9 | 24.2 | 22.9 | 20.5 |
| $w_{sat}$ [m³ m⁻³] | 0.468 | 0.482 | 0.477 | 0.476 | 0.452 | 0.468 | 0.47 | 0.472 |
| $w_{wilt}$ [m³ m⁻³] | 0.143 | 0.154 | 0.139 | 0.146 | 0.11 | 0.134 | 0.144 | 0.149 |
| $w_{fc}$ [m³ m⁻³] | 0.229 | 0.24 | 0.225 | 0.231 | 0.19 | 0.218 | 0.229 | 0.2 |
| $K_{sat}$ [mm h⁻¹] | 252.2 | 79.2 | 59.1 | 97.7 | 127.7 | 20.2 | 185.4 | - |

and,

$$w_{fc} = 89.0497 \cdot 10^{-3} C^{0.3495}. \tag{6}$$

The soil parameters are computed for each depth and then averaged on the soil vertically to get one value for each of the 9 sampling points. To consider commensurate variables, the Relative Water Content $W_r$ [-], was used for each soil layer:

$$W_r = \frac{W - w_{wilt}}{w_{sat} - w_{wilt}}, \tag{7}$$

where $W$ [m³ m⁻³] is the measured soil water content. A $W_r$ equal to 1 means that the saturation of the soil layer is reached, while 0 means that its wilting point is reached.



## 3.6 Infiltrometry test

The Beerkan infiltration method was used to determine the saturated hydraulic conductivity at the sampling points. This method
is detailed by Haverkamp et al. (1994) and Braud et al. (2005). A PVC cylinder of diameter 25 cm was used with 1 L of water
poured at each iteration. For estimating retention curves and hydraulic conductivities, the BEST (Beerkan Estimation of Soil
Transfer parameters) method (Lassabatère et al., 2006) was applied. BEST is approaching the series of cumulative infiltration
rates and instantaneous infiltration rates by the expressions provided by Haverkamp et al. (1994), which involves the sorptivity
and the hydraulic conductivity at saturation $K_{sat}$ of the soil (Van Genuchten, 1980; Burdine, 1953; Brooks and Corey, 1966).
These expressions involve three parameters of form which are determined from particle size distributions (Fuentes et al., 2017)
and based on capillarity models (Haverkamp et al., 1994).

## 4 Results

### 4.1 Snow depth estimation

The snow depth estimations from the two SnowTrees at LaChaux and Chalet are presented in Fig. 2, together with the visual
snow depth estimation at the Auberge station. The dynamics of the snow depths are consistent between the temperature-
based and visual methods. For example, the snow depth peak on 21 January 2023 is represented with the same timing in the
SnowTree and the visual scale results. The SnowTree better monitors the snow depth when it is well exposed to direct solar
radiation, i.e., after mid-January (time-lapse images showed undetected frosted residual snow remaining along the wood mast).
In a narrow and shadowed valley like the Nant Valley, the diurnal heating due to solar radiation when the sun position is
low in winter is not dramatically varying and the determination of snow depth based on logger temperature is challenging. A
visualisation of the solar illumination in the valley, including hill-shading effects as computed with the *hillshading* function
(*insol* R package; Corripio and Corripio, 2019) is plotted in Fig. 2. Inconsistent values due to these effects are manually
removed but weaker perturbations may remain in the temperature measurements during cold days. This issue is tackled by
applying different thresholds whether the daily temperature exceeds 1°C. This differentiated threshold, combined with manual
checking of the images from the camera, makes SnowTree a potentially valuable instrument for assessing snow depth in remote
areas.

### 4.2 Snow Water Equivalent estimation

The 12 h averaged $S_{WE}$ computed with the Cosmic Ray Sensor and with the Snow Melt Analyzer are presented in Fig. 3,
together with the hourly total precipitation and temperature recorded at the Auberge weather station. Despite the time series
being short (the CRS instrument stopped due to an unstable electric feeding in early February 2023), different processes were
identified: (i) regarding the snow accumulation periods, an increasing snow depth and increasing $S_{WE}$ were recorded without
considerable increasing in $L_{WC}^{bottom}$ when precipitation occurs with $T_{air} < 1\ °C$. This can be seen, for example, between 8 and
18 December, 2022; (ii) an increase in snowpack total liquid water content caused by liquid precipitation (i.e., $T_{air} > 1\ °C$),



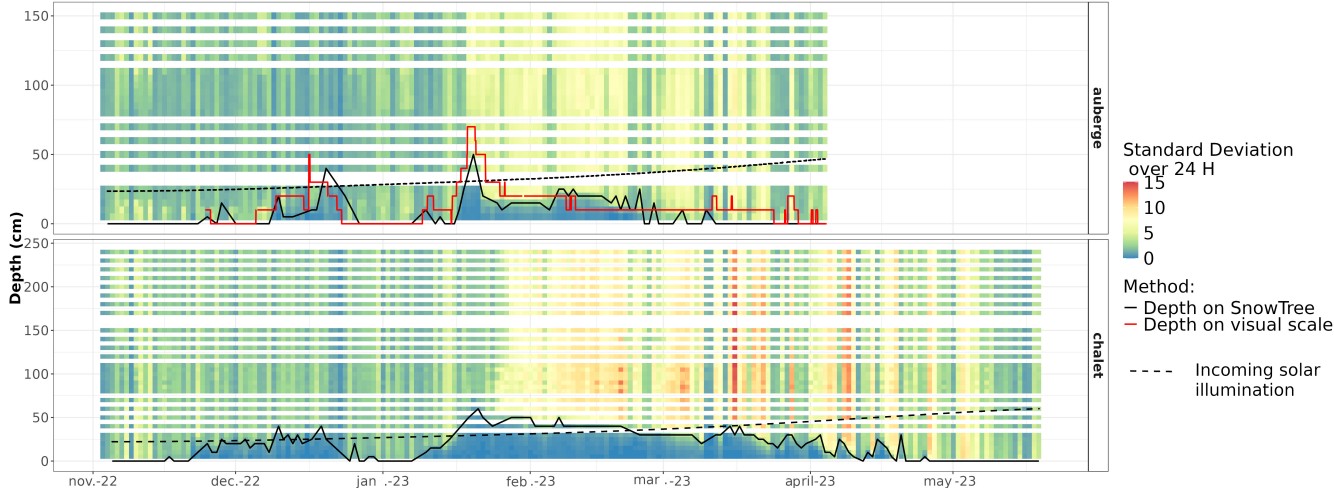

**Figure 2.** Temperature standard deviation computed over 24 h ($24hSTD$) recorded by the iButton loggers on the SnowTree instrument at the Auberge and Chalet locations (background colors). Black and red solid lines represent the snow depths obtained from the SnowTrees and the visual scales, respectively. The dashed line corresponds to the intensity of solar illumination received at the measurement points.

with an increase both in $S_{WE}$ and in $L_{WC}^{bottom}$, without significant decrease in snow depth; and (iii) an anomaly in snowmelt flux when $T_{air} > 1\,°C$ and when $L_{WC}^{bottom}$ behaves oppositely to the variation of the $S_{WE}^{CRS}$.

This shows that the variation of $S_{WE}$ does not necessarily correctly represent the melt rate, as it is often considered in other studies (Kampf et al., 2015; Fang et al., 2019; Archer and Stewart, 1995). The vertical heterogeneity of the snowpack influences the water content within the snow layer. By focusing on the bottom snow layer, our novel approach isolates actual melt from general changes in snow depth and $S_{WE}$, allowing for more accurate local melt rate estimations, which is a key variable of interest. In addition, the only punctual snow density measurement realized within the CRS recording period (recorded on 5 December, 2022) gave a snow density of $\delta_{obs} = 293.7 g.dm^{-3}$ and a 24 cm snow depth. For this time point, the CRS provided a $S_{WE}$ of 7.92 cm, what is equivalent to 26.9 cm using the $\delta_{obs}$ density. This shows that the estimation of $S_{WE}$ by the CRS led to an estimation of snow depth consistent with the measured snow depth (26.9 cm against observation of 24 cm).

### 4.3 Soil parameters estimation

The clay, sand, and silt fractions resulting from the laser granulometry analysis, together with the water content at different phases computed through the Clapp and Hornberger (1978)'s equation, are summarised in Table 1. Using the USDA soil texture classification, all the samples are in the Silty Loam category, which is consistent with moraine silty deposits. In addition, the results are consistent with previous studies in the Nant Valley (Grand et al., 2016; Cianfrani et al., 2019). The hydraulic conductivity at saturation ($K_{sat}$) and the average soil water content are also presented in Table 1. The values of $K_{sat}$ obtained are consistent with general values considered by Cosby et al. (1984) for soil classes. Cattle trampling and grazing affects $K_{sat}$;



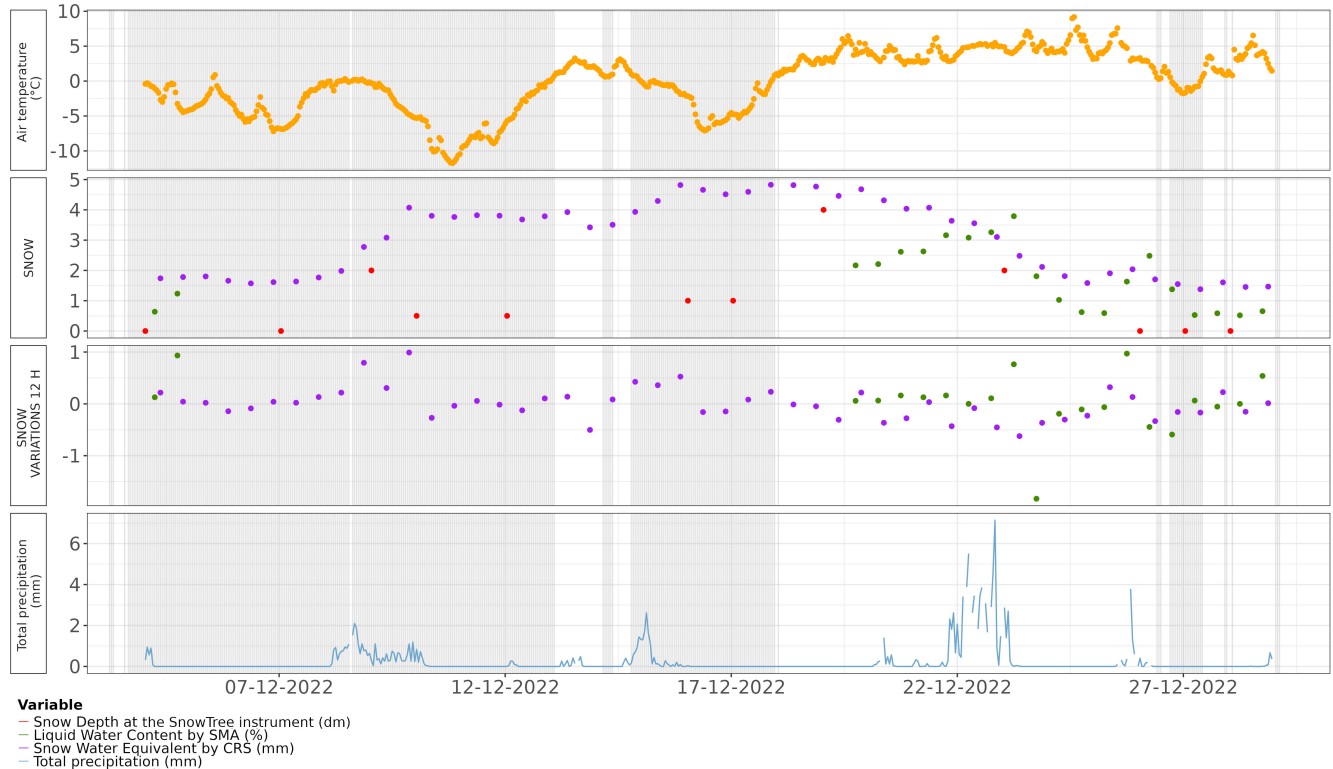

**Figure 3.** Snow Water Equivalent computed from Cosmic Ray Sensor measurements $S_{WE}^{CRS}$ [mm], and its hourly variations averaged over 12 hours $\Delta S_{WE}^{CRS}$ [mm], together with the liquid water content of the bottom snow layer measured by the Snow Melt Analyzer $L_{WC}^{SMA}$ [%] and its hourly variations averaged over 12 hours $\Delta L_{WC}^{SMA}$ [%]. The grey bands indicate the time step with temperatures below $1°C$.

non-grazed locations (Combe, Protegee, Auberge) present $K_{sat}$ considerably higher than grazed locations (Chalet, LaChaux, Pissenlit). In addition, a dense grass root system is observed in Combe, which might act to reduce the infiltration rate at this location.

## 4.4 Inter-annual variability of the snow contribution

The three monitored winters markedly differ in terms of snowpack dynamics as indicated by e.g., the climatic and hydrological variables observed at the outlet of the catchment (Fig. 4). The Winter 2021–2022 consisted of an intense melting period in early winter, the 2022-2023 winter presented chaotic snow conditions, with the alternation of accumulation and melting periods, and the 2023-2024 winter exhibited a shorter snow cover period, but with a melt period concentrated in the spring season. For the 2021-2022 winter, the first snow accumulation period occurred in early winter (December), followed by a warmer melting

period in January, another accumulation period in late winter (March), and a gradual melting in April. For the 2022-2023 winter, the snow season consisted mainly of an alternation of accumulation and fast melt periods, due to the sharp increase in



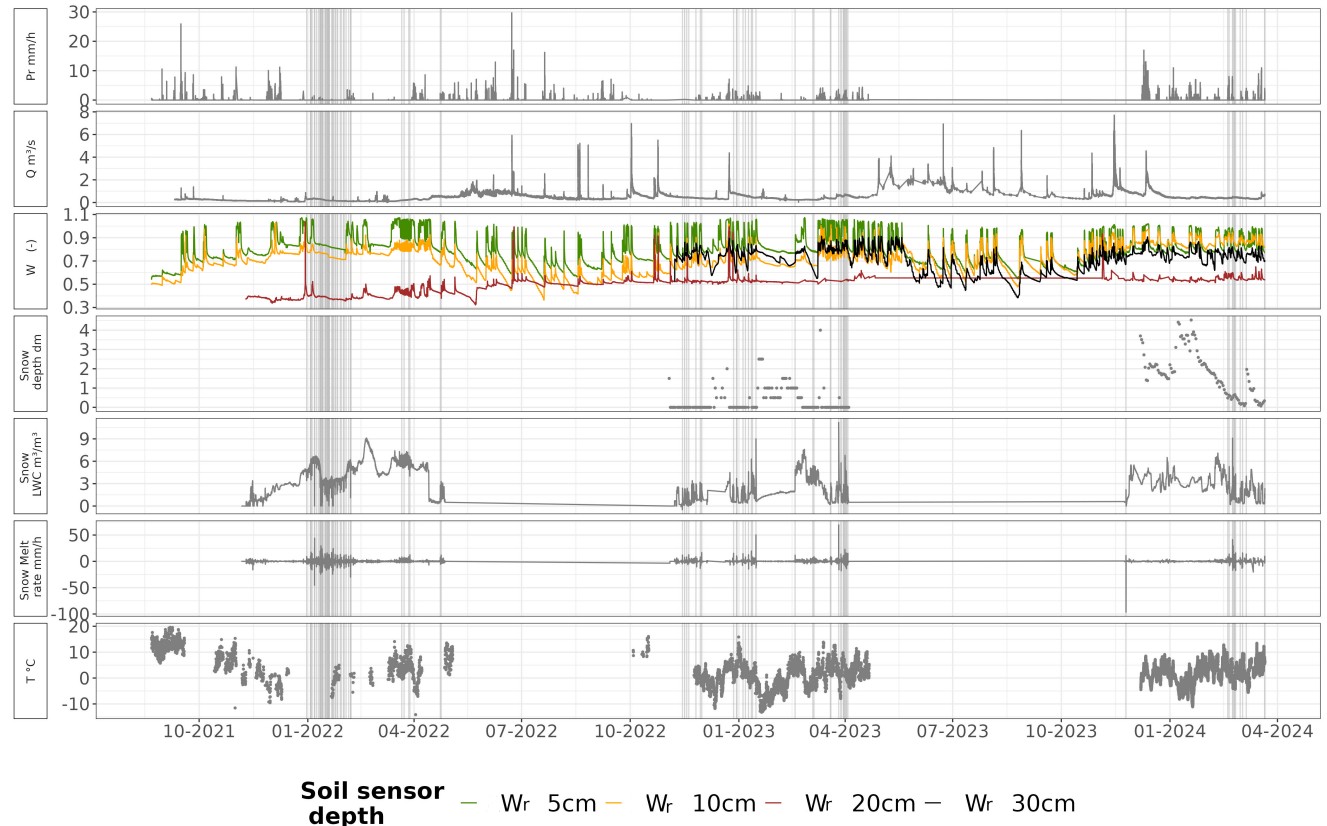

**Figure 4.** Soil moisture (HU), Precipitation (PR), Temperature, discharge at the outlet (Q), Water content of the bottom snow layer and snow depth measured at the Auberge point, together with $K_{sat}$ values and zoom into periods of interest, for the entire recording period: from September 2021 to March 2024. The vertical grey lines represent the time steps when the hourly melt rate overpasses the $K_{sat}$ value measured at the Auberge soil sampling point.

temperature. The spring melt period expands from mid-February to the beginning of April. Over the entire period, at the three measurement points and at all of the instrumented depths, none of the recorded hourly soil temperature reached values lower than 0.5°C, meaning that soils do not get frozen. The annual cycle is clearly observable for the upper soil layer, i.e., the 10 cm and 20 cm depths, with a progressive decrease after the snowpack disappearance in spring (Fig. 4). Details on this dynamic are discussed next.

## 4.5 Dynamics of the infiltration-runoff partition

The melt rate intensity is generally lower than the conductivity at saturation determined from the infiltrometry tests, indicating that infiltration-excess surface saturation is unlikely to occur.(Fig. 4). During the monitoring period, melt rates exceeded $K_{sat}$ mainly during the winter melt in 2022 (January and February) and during the spring melt in 2023 and 2024 (Mars and April).




During winter, when peaks of melt are concomitant with liquid precipitation, they lead to a quick response of soil moisture and stream discharge. As illustrated in Fig. 5a, when they are not associated with precipitation, winter peaks of snow melt do not lead to a significant response of discharge or soil water content. Therefore, since infiltration excess overland flow are non-majority of the contribution during winter, the discharge response must be driven by shallow subsurface stormflow in winter.
This nicely emphasis the phenomenon known as the "inverse storage effect" (Ceperley et al., 2020; Fang et al., 2019; Benettin et al., 2017).

During spring, the peaks of snowmelt that exceed $K_{sat}$ are mainly associated with changes in the energy budget, led by diurnal variation of radiation and temperature. Fig. 5b clearly illustrates these diurnal cycles. In this example, diurnal peaks of snowmelt that exceed $K_{sat}$ (or closely) happen whereas the superficial top layers are already close to saturation ($W_r$ higher
than 80%). The response time of both the two superficial soil layers and of the discharge is very fast, about 3-4 h. The response of the third layer is slower (5-6 h) and attenuated. In addition, the melt peaks are more often associated with mixed (rain and snow) precipitation. Combined with the low spring snow depths (Fig. 2), precipitation acts to strongly accelerate spring snow pack melt. In addition, the discharge measurements are taken significantly downstream of the monitoring sites, highlighting the critical examination of how these measurement points in the catchment connect to the location where discharge is observed

## 4.6 Influence of snow cover on soil layers responses

As shown in Fig. 4, the occurrence of snow cover modifies the response of the soil water content to incident flux (melting or precipitation) for the two superficial layers (up to 20 cm depth). Table 2 synthesizes the characteristics of the three considered soil layers with or without snow cover. The soil relative water content averaged over each period is slightly higher when the soil is covered by snow than when it is not covered. For the two superficial layers, periods with snow cover are characterized by
longer periods of saturation, with numerous occurrences of periods of 6 consecutive hours with $W_r$ higher than 90% ($W_r^{6H90}$, see Table 2). On the contrary, the 30 cm depth layer does not strongly react to the presence of snow cover, neither regarding the $W_r^{6H90}$ nor the average $W_r$. In addition, in both cases of winter (Fig. 5a) and spring (Fig.5b and 5c) melting, the response of the 30-cm layer to snowmelt peaks is delayed and attenuated compared to the two superficial ones. In some cases, this deep layer can even present no reaction to the melt flux and the saturation for this layer is rarely reached. Conversely, as shown in Fig. 5d,
the response of soil to summer liquid precipitation (without snow cover) consists of fast responses to the peak of precipitation (about 2 h) and progressive emptying. The three considered soil depths have similar response times to precipitation peaks, but the two superficial layers present faster-decreasing rates than the 30 cm layer one.

These observations lead to the conclusion that, despite the intensity of the melt rate being lower than the conductivity at saturation, the infiltrated volumes hardly reach the soil layer below 30 cm at this measurement point. The main increase of the
soil water content of this soil layer mainly coincides with liquid precipitation in late spring. The transfer of infiltrated melted snow mainly takes place in the superficial soil layer. These results suggest that lateral transfer in superficial soil layers (above 30 cm depth) might be the main process leading snowmelt flux toward the river network. These results corroborated the results of Ceperley et al. (2020), who found that young water is the main contributor to discharge during winter. Kampf et al. (2015) also noticed empirical evidence of lateral transfers during spring melt, based on audible sounds.





**Table 2.** Characteristics of the three considered soil layers, when soil is and is not covered by snow: average $W_r$ and occurrences of periods of 6 consecutive hours with $W_r$ higher than $90\%(W_r^{6H90})$, on average for the three monitored winters.

|  | 414 days with snow cover | | | 438 days without snow cover | | |
|---|---|---|---|---|---|---|
|  | L1 (10 cm) | L2 (20 cm) | L3 (30 cm) | L1 (10 cm) | L2 (20 cm) | L3 (30 cm) |
| Average $W_r$ [-] | 0.873 | 0.779 | 0.476 | 0.753 | 0.654 | 0.500 |
| $W_r^{6H90}$ [d] | 546 | 127 | 3 | 219 | 31 | 1 |

## 4.7 Spatial variations of the snowmelt-infiltration processes

Last, we explore the spatial variability of snowmelt-infiltration dynamics, focusing on the hydrological response at the 25 cm depth (Fig. 6). The annual dynamics of the soil moisture are similar at the three measurement points. Overall soil moisture increases with increased snow pack; this is specifically noticeable in December 2022. Diurnal cycles of soil moisture occur during the spring melt season, from April to May 2022 and from mid-February to late April 2023. The dynamics of soil moisture during spring melt are shifted at the three sites, due to the difference in altitude. However, the overall spring melt periods coincide. In spring, the peaks of soil moisture due to diurnal melt rates as described earlier, are shifted from about five hours between Chalet and Auberge stations. On the other side, the peaks due to liquid to mixed precipitation in winter, as well as liquid precipitation in summer are shifted by only two hours. This is due to the lag between diurnal snowmelt at different altitudes, whereas precipitation happens more simultaneously in the catchment. The soil moisture peaks are less pronounced at La Chaux, compared to the two other ones. This can be since the soil conductivity at saturation is smaller at La Chaux than at Chalet of Auberge, in particular, due to the impact of cow grazing. Another reason can be related to the fact that the snowmelt rate can be slower with increasing elevations, with a smaller diurnal amplitude of air temperature.

The correspondence of the soil moisture time series at these three measurement sites, together with the relative similarity of the $K_{sat}$ values at measurement points presenting similar soil contexts (see section 4.3), allows to extent the results obtained at the Auberge sites. Processes at the Chalet, Petit Pont, Pissenlit, and La Chaux points may then correspond to the processes observed at the Auberge point: these locations may act mainly as areas of lateral transfer in superficial soil layer (less than 30 cm depth) during spring melt periods but also during winter accumulation periods. However, punctual areas with higher values of $K_{sat}$, illustrated by the Combe point, may act as areas of recharge, with more important vertical infiltration rates. This emphasizes the crucial role of these non-pastured, protected areas in mountains.

## 5 Discussion

One of projected impact of global increase in temperature in alpine area is an increase in the intensity of the winter melt rates and an increase in intensity and volumes of spring melt (Han et al., 2024; Masson-Delmotte et al., 2021). This intense melt period, combined with projected decrease of seasonal snow volumes, is also projected to cause a shift in the snow melt period of about one month at the horizon of 100 years (Hock et al., 2019), that is, for the studied area, from March-June to



February-May (Antoniazza, 2023). In this context, the rapid infiltration in the upper soil layers and the fast response of the river flow highlighted in this work corroborates the increase in the flood risk during the Spring season Kundzewicz et al. (2014). In addition, the extend of the low flow period (from July-August to June-August) could lead to an increasing risk of damage to the riverine ecosystems (Pletterbauer et al., 2018) and to a deficit in the seasonal feeding for hydroelectricity.

Even though this work is limited by compiling only three years of data, which occasionally includes significant gaps and
restricted recording periods, with limited measurement points and uncertainties, especially regarding weather and snow depth data, it nonetheless provides an uncommon combination of soil and snow data using various acquisition methods. The resulting dataset represents a valuable contribution to the understanding of mountainous environments.

Moreover, the response of deeper soils and groundwater is not considered here. In particular, the flows and storage in unconsolidated subsoils such as moraine deposits of landslides and rockfalls can be prone to be important. This work paves
the way for a better understanding of the contribution of snow melt to deep water recharge in mountain : at the study site, the superficial soil layers (i.e. above 30 cm) appear to limit the infiltration flux toward deeper zones. The assumption can the be done that the recharge of deep storage might then preferentially occur over specific areas presenting no or little developed soils, such as bare rock or uncovered moraines. This assumption would need to be further explored by future hydrogeology studies. Piezometers were installed at different locations in the Nant valley by Thornton (2020), however the recording but
were no longer recording during this study. Further instrumental works could allow to better describe the circulation below 30 cm depth.

In addition to its contribution to process understanding, the snow-and-soil records can be used as validation or calibration data to improve process simulation in physically-based hydrological models, in particular for local scale studies. These observations could assist in refining aspects physically based surface schemes implemented at the measurement point (Thornton
et al., 2022).





(a) Example 1: Early winter 2021

(b) Example 2: Diurnal melt during Spring 2022

(c) Example 3: Spring 2024

(d) Example 4: Summer 2023

**Figure 5.** Zoom into specific periods for the variables Soil moisture (HU), Precipitation (PR), Temperature, discharge at the outlet (Q), Water content of the bottom snow layer and snow depth measured at the Auberge point.



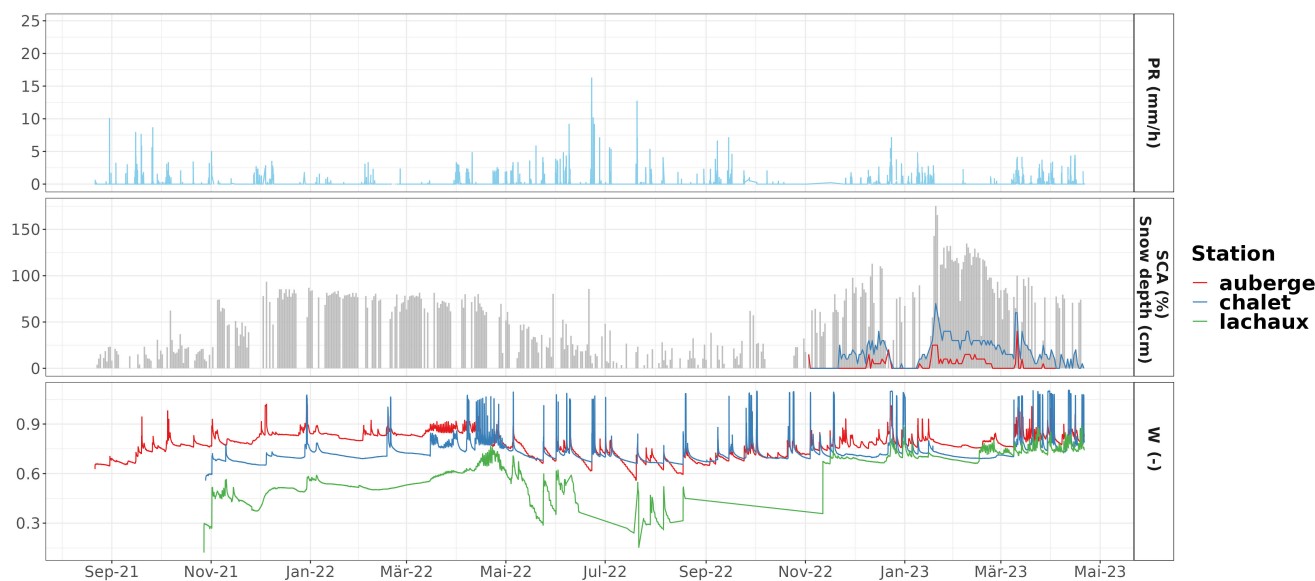

**Figure 6.** Soil moisture measured at 25 cm depth at the three measurement points, together with precipitation recorded at Auberge station, the MOD10A1 daily values of Snow Cover Area, on average over the entire catchment and the snow depths estimated at the SnowTree instruments at Auberge and Chalet spots.



## 6   Conclusions

We combined innovative tools to locally monitor both the snowmelt rate and the response of the soil water content to it. An uncommon method was tested to assess snow depth in the absence of a measuring station in remote areas: the snow depth is estimated based on the temperature measurements at tiny sensors along a wood mast. Two approaches are crossed to estimate
the snowmelt rate: the measurement of the dielectric constant of the bottom snow layer and the measurement of the total snow water content through a cosmic ray sensor.

The main conclusions of our instrumental experiment are as follows:

– Even if the measured snowmelt rate is generally lower than the soil conductivity at saturation, the response time of superficial soil layers above 20 cm and discharge to melt events is relatively fast, between 2 and 5 hours. No long-term
340        storage is observed in these superficial soil layers.

– An inverse storage effect is demonstrated during winter: although snowmelt events in winter lead to rapid stream discharge responses but of low amplitude, the surface water content of the soil decreases over the periods of snowpack accumulation. This corroborates the fact that young waters actively contribute to discharges in winter.

– The water content of the soil at 30 cm depth is little influenced by the melt flux, or exceptionally influenced during melt
345        events coupled to precipitation. On the opposite, this soil layer is fast reacting to summer precipitation.

– Soil moisture recorded at different locations present similar temporal dynamics. The processes observed at the point measurement can thus be extrapolated and areas of superficial runoff generation can be described. On the other hand, less commune areas potentially allowing more important vertical infiltration may be assumed.

This work emphasizes that the fast response of soils and discharge to snowmelt increases the risk of low flow and water
shortage outside of the melt period. This highlights the dependency of the water availability on the delayed melt of the snow pack at high elevations. Finally, interactions between the fast-reacting unsaturated zone of the soil and the snow-groundwater at the catchment scale must be considered in future work.

*Data availability.*   All the data produced in this work are freely available through the dedicated Zenodo platform: zenodo.org/communities/vdn/. Soil moisture, soil temperature data and granulometry results can be download at zenodo.org/records/10136586 (Eeckman, 2023). SMA snow
melt rate and snow water content data, as well as SnowTree data can be download at zenodo.org/records/11580271 ,(Eeckman, 2024).

*Author contributions.*   JE initiated and developed the research project, raised funding for the instrumentation and fieldwork, carried out the fieldwork and data analysis, as well as the writing of the paper. NP advised the writing of the paper and significantly contributed to its redaction. BD actively assisted with the fieldwork and participated in the processing of field data. FM actively assisted with the fieldwork and preparation of materials. JT installed the CRS instrument on the field and contributed to the analysis of CRS data.



*Competing interests.* NP is a member of the editorial board of Hydrology and Earth System Sciences.

*Acknowledgements.* The SMA and SnowTree instruments were financed thanks to the investment fund (FINV) of the University of Lausanne. The field assistance was financed thanks to the contribution of the UNIL Institute of Geography and Sustainability (IGD). Other costs related to fieldwork were financed by the IGD fund for fieldwork. The authors would like to thank the director of the IGD for this support. NP was supported by the Swiss National Science Foundation (SNSF), Grant 194649 ("Rainfall and floods in future cities").



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
