# Peer review of "Multi-instrumental monitoring of snowmelt infiltration in Vallon de Nant, Swiss Alps"

_EGUsphere, 2024_

## Referee Comment (RC3)

[referee-annotated manuscript omitted]

---

## Author Response (AR1)

**Authors' Response to Reviewer Comments N°1**

We sincerely thank the reviewer for their thorough and constructive feedback, which has greatly helped us improve the clarity, organization, and scientific rigor of our manuscript. Below, we provide detailed responses to the points raised and outline the revisions made in the manuscript.

- **Reorganization of the paper**
  We have revised the structure of the Introduction, Results, Discussion, and Conclusions sections to better align with the research questions and ensure a logical flow of information.

  The **Introduction** has been revised to clearly present the three research questions and their significance. The structure of the introduction is now :

  1. State of the art for soil processes understanding in mountain.

  2. State of the art for snowmelt assessment : modeling method and the different instrumental method. The advantages and drawbacks of each method is presented.

  3. Lack of knowledge in understanding in processes of snowmelt infiltration over alpine developed soils. Specifically, the following sentences are added *: 'The current understanding of the physical processes remains then insufficient to explain the role of the unsaturated zone for the infiltration of snowmelt in mountains. In particular, additional descriptions of the partition between surface runoff and vertical infiltration of snowmelt rate into soil layers, the depth reached by vertical percolation and the depth of preferential circulations for lateral transmission along the slopes, and the response times of both the soil moisture and the river discharge, are needed.';*

  4. Aim of the paper. The following sentence is added : *'This work aims to address the two issues of i) providing an accurate method for the estimation of snowmelt rate, and ii) accurately detailing the physical processes involved in the unsaturated zone during the infiltration of snowmelt at a study point.'*

  5. Overall methods : instrumental approach for snow and soil processes, 3 years data set, …

  The **Results** section is now organized into three subsections corresponding to the research questions:
  1. **Multi-instrumental estimation of snowmelt rate.**
     The results of each of the 3 snow instrument (SnowTree, CRS, SMA) are presented and compared.

  2. **Partitioning of infiltration and runoff during snowmelt.**
     The intensities of melt rates are compared with the infiltration capacity of soils to better quantify the partition between surface runoff and infiltration of the melt flux, in winter and in spring season.

  3. **Vertical and lateral transfers of meltwater and its impact on streamflow.**
     The vertical percolation of the melt flux into soil layers and its response to melt events are described. The typical response times of the soil layers at the

three instrumented sites in the catchment to melt event are described. The response time of river discharge is also presented.

- The **Discussion** section revisits each research question, providing detailed interpretations of the results and connecting them to broader implications. The paragraphs that were previously mixed into the Results section (e.g., lines 216–223) have been relocated and expanded within the Discussion.

  The Discussion section is now organized as follow :
  - 1. **Uncertainties in snowmelt estimation**
    The limit of each snow instrument is presented, and their complementarity for snowmelt assement is presented. In particular, the impacts of rain-on-snow phenomenon is investigated.

  - 2. **Inter-annual variability of the snow contribution**
    The variability between the 3 recorded years is described. Divergences and common behaviors within the recorded seasons are drawn. A open discussion about the potential impat of climate change on the presented results if raised.

  - 3. **Link with undeground water storage**
    The issue of the interaction with deep storage is detailed. A discussion about the possible futur works about groundwater investigation in the catchment is open.

- The **Conclusions** is restructured to explicitly summarize the answers to the research questions and highlight the contributions of the study. The Conclusion section is now organized as follow :
  - 1. Reminder of the objectives of the paper and the three main points of the results
  - 2. A quick summary of the main results, organized along the same structure than the Result section (3 bullet points)
  - 3. The possible application of this work as validation/calibration data for futur modeling works is highlighted.

- **Clarification and Consistency of Variables**
  We agree that the use and explanation of variables needed improvement. The following revisions have been made:
  - Precipitation is now consistently denoted as $P$ (mm·s⁻¹).
  - The variable HU has been replaced by $W_r$ (m³·m⁻³) throughout the text, figures, and captions.
  - The liquid water content of the bottom snow layer, previously $L_{bottom\_WC}$, is now denoted as $\Lambda$ (%).
  - Temporary variables such as $W_r6H90$, $\Delta SWE$, and $\Delta LSM$ have been removed for clarity.
- **Improvements to Figures and Captions**
  We have made substantial changes to the figures and captions:

- **Figure captions** now describe the key findings and are more detailed. Unnecessary references to "Figure X presents..." have been removed, as suggested.
- The **quality and readability** of figures have been improved:
    1. The figures have been resized to ensure clarity, especially Figures 4 and 5, where narrow plots made interpretation difficult.
    2. Line thicknesses and colors have been adjusted to enhance distinction (e.g., soil sensor depths in Figure 4).
    3. Precipitation is now presented as a bar chart (hyetograph) in line with conventions.
- Sub-figure labels (a, b, c, etc.) have been added where appropriate, and these labels are referenced in the captions for clarity.
- Figure 6, values of snow cover area (SCA) exceeding 100% are due to the spatial resolution of the gridded MODIS product. This is specified in the legend of Figure 6.

**Specific Responses to Comments**

- **Introduction (lines 47–49):**
Many factors can affect the vertical heterogeneity of the snowpack: transport by wind (addition or removal), snow sublimation, and transformations of the snow at the surface or within the snowpack. The vertical heterogeneity of snow (particularly the density and ice content of each layer) may strongly affect the vertical percolation of meltwater in the snowpack. In particular, icy layers can block vertical transfer. The intercomparison of the CRS, the SnowTree, and the SMA results allows investigation into the influence of the vertical heterogeneity of snow on the estimation of the melt flux. The CRS (respectively, SnowTree) instrument considers the SWE (respectively, depth) of the whole snowpack, without regard to whether the wet snow is located at the top or the bottom of the snowpack (i.e., no consideration of vertical heterogeneity). Conversely, the SMA considers only the liquid content of the bottom snow layer. The assumption made in this work is that the liquid content of the bottom snow layer is bound to percolate toward the soil by gravity and will not undergo further transformation. Even though a slight delay may be induced by gravity transport through the bottom layer, the liquid content of the bottom snow layer appears to be an accurate representation of the snowmelt flux.

- **SMA Instrument:**
While we acknowledge the reviewer's concern about the lack of independent validation for the SMA, we note that our study's results corroborate the SMA data with other measurements (e.g., visual snow depth and cosmic ray sensor data). This comparison helps build confidence in the instrument's reliability.

- **Visual Scale at the Auberge site:**
The visual scale at the Auberge site was of great importance for calibrating the SnowTree instrument. Despite that its vertical resolution is low, an that is remains a non-automatic assessment of the snow depth (visually estimated on each picture), it remains a very human checking of the data obtained but automatic sensors.

- **Formatting and Style Issues:**

    - Page 1, line 2: We have replaced "parameters" with "variables" for accuracy.
    - Figure 1: The element D is now described in the caption.
    - Line 135–136: The notation for SWE has been corrected to uppercase without subscripts.

- Line 148: Redundant text has been removed.
- Line 148: The diameter of the density measurement cylinder was 12cm. This is added in the text.
- Line 221: The symbol for density is now ρ, consistent with snow hydrology conventions, and units are reported in $kg \cdot m^{-3}$. The precision of density values has been reduced to one decimal place.

- The statement about the SMA method being "uncommon" has been reworded to reflect its novel application in this specific context, acknowledging prior use in the literature. However, the multi-instrumental estimation of snowmelt proposed in this work (SMA, CRS, SnowTree and SnowScale), remain unusual in literature and represent an actual scientific breakthrough to provide a robust method for estimating snowmelt.

**Authors' Response to Reviewer Comments 2**

We thank the reviewer for their detailed and constructive feedback, which has greatly contributed to improving the quality and clarity of our manuscript. Below are our point-by-point responses to your observations.

**Main comments:**

**Comment :**
First, it is unclear whether the primary focus is the proposed instrumentation approach or the processes occurring in the study area.

**Response :**

Finding a suitable and robust method to measure snowmelt was necessary to perform the analysis of the process. The authors recognize that this dual scope does not meet typical standards for article structure, but both aspects of research were necessary to complete the study. The introduction is deeply restructured to better explain the approaches and the objective of the work. In particular, the following sentence is added: "*This work aims to address the following two issues: i) provide a precise method for estimating the snowmelt rate, and ii) precisely detail the physical processes involved in the area not saturated during snowmelt infiltration. at a study point.*

**Comment :**
The state of the art and how this research aims to address existing knowledge gaps are also not sufficiently explained. Additionally, the research questions lack clarity, and the presented results do not provide a strong foundation for conclusive statements.

**Response :**

We acknowledge the need for a more coherent structure throughout the manuscript. The structure of the Introduction, as well as the structure of the Results and Discussion sections and the Conclusion are revised.

The **Introduction** has been revised to clearly present the three research questions and their significance. The structure of the introduction is now :
> 1. State of the art for soil processes understanding in mountain.
> 2. State of the art for snowmelt assessment : modeling method and the different instrumental methods. The advantages and drawbacks of each method is presented.
> 3. Lack of knowledge in understanding in processes of snowmelt infiltration over alpine developed soils. Specifically, the following sentences are added *: 'The current understanding of the physical processes remains then insufficient to explain the role of the unsaturated zone for the infiltration of snowmelt in mountains. In particular, additional descriptions of the partition between surface runoff and vertical infiltration of snowmelt rate into soil layers, the depth reached by vertical percolation and the depth of preferential circulations for lateral transmission along the slopes, and the response times of both the soil moisture and the river discharge, are needed.';*
> 4. Aim of the paper. The following sentence is added : *'This work aims to address the two issues of i) providing an accurate method for the estimation of snowmelt rate, and ii) accurately detailing the physical processes involved in the unsaturated zone during the infiltration of snowmelt at a study point.'*
> 5. Overall methods : instrumental approach for snow and soil processes, 3 years data set, …

> The **Results** section is now organized into three subsections corresponding to the research questions:
> 1. **Multi-instrumental estimation of snowmelt rate.**
>    The results of each of the 3 snow instrument (SnowTree, CRS, SMA) are presented and compared.

2. **Partitioning of infiltration and runoff during snowmelt.**
   The intensities of melt rates are compared with the infiltration capacity of soils to better quantify the partition between surface runoff and infiltration of the melt flux, in winter and in spring season.

3. **Vertical and lateral transfers of meltwater and its impact on streamflow.**
   The vertical percolation of the melt flux into soil layers and its response to melt events are described. The typical response times of the soil layers at the three instrumented sites in the catchment to melt event are described. The response time of river discharge is also presented.

**Comment:**
To assess the broader applicability of the proposed instrumentation methodology in other mountainous regions worldwide, it is crucial to include a discussion about its potential scope and limitations. The extent to which the assumptions made condition the results being presented and opportunities for improvement should be included in the analysis and discussion. […]
Finally, the discussion should align more closely with supported findings, emphasizing the study's unique contributions over literature-based insights, and rephrasing conclusions to highlight original findings.

**Reponse :**
The **Discussion** section is strongly restructured to revisit each section of the Results section. And to discuss the limits and the potential application of each obtained result.  The Discussion section is now organized  as follow :
1. **Uncertainties in snowmelt estimation**
The limit of each snow instrument is presented, and their complementarity for snowmelt assement is presented. In particular, the impacts of rain-on-snow phenomenon is investigated.

2. **Inter-annual variability of the snow contribution**
The variability between the 3 recorded years is described. Divergences and common behaviors within the recorded seasons are drawn. A open discussion about the potential impat of climate change on the presented results is raised.

3. **Link with undeground water storage**
The issue of the interaction with deep storage is detailed. A discussion about the possible futur works about groundwater investigation in the catchment is open.

The **Conclusions** is also restructured to explicitly summarize the answers to the research questions and to highlight the contributions of the study. The Conclusion section is now organized as follow :
1. Reminder of the objectives of the paper and the three main points of the results
2. A quick summary of the main results, organized along the same structure than the Result section **(3 bullet points)**
3. Highlight of the novelty of the work
4.The possible application of this work as validation/calibration data for futur modeling works is highlighted.

**Comment :**
Revising figures for clarity, adding quantitative evidence for statements, and analyzing lag times between precipitation, streamflow, and soil moisture would enhance interpretability.

**Response :**
In accordance with the different reviews, the appearance of all figures has been greatly improved. The legends and axis legends are enriched and the order of the subfigures is reorganized for better clarity. The issue of response time of soil layers and flow rate is specifically addressed in the third paragraph of the Results section. The time periods shown in Figure 5 are chosen to better represent the response time to merger events.

**Major revisions:**

**Comment :**
The first statement (L13:L14) forces the introduction of the concept of snowmelt infiltration. Is the important role of the snow due to snow infiltration or due to its contribution to surface runoff? Is the role of infiltration clear enough to make this statement? I suggest you rephrase the first sentences to emphasize the well-known role of snow (e.g., natural reservoir of fresh water), then the processes involved in snowpack changes, and conclude with the idea of the partitioning of surface runoff and infiltration and its importance in water availability.

**Response :**
The introduction is completely restructured (see above for the new structure). This first sentence is then deleted and the introduction begins with the following sentence: *"Understanding the processes controlling the infiltration of snowmelt flux into alpine soils remains one of the difficult questions in hydrology. »* Sentences regarding the predicted impacts of climate change on snowmelt are moved to the Discussion section.

**Comment :**
When you mention "... changing the snowmelt infiltration dynamics" (L16:L17) you are hiding all the impacts that an earlier melting would have. What about seasonality, peaks, among others? Logically many things will change, but highlighting the dynamics does not seem intuitive to me because, according to what has been mentioned up to the point of this sentence, there is also no information regarding the role of this process, or is there? I suggest you rewrite the paragraph to give more support to this idea (if there is evidence).

**Response :**
Indeed, the consequences of climate change on snowmelt are very diverse. This work focuses on one of these impacts described in the 6th IPCC report for the Alps: the intensification of melting rates due to the increase in temperature, particularly in spring. The results on the infiltration capacity of soils provided in this work raise the question of whether higher melting rates could be infiltrated into soils (and to what depth?) or whether this would induce more surface runoff. An important contribution of this article is to show that the infiltration capacity of soils is not often exceeded by melt rates, except during spring melt peaks. This work makes it possible to discriminate the range of melting intensities that may or may not be infiltrated into the soil. This topic is covered in the second paragraph of the Discussion section. This is also mentioned in the Conclusion.

**Comment :**
When citing studies that have been applied in different regions and/or catchments, provide some examples and, perhaps, some of the important findings of such research. This will allow you to establish a certain relationship between your results and those obtained in similar areas. This could be useful to include in the discussion which, currently, is more focused on hypothesizing than discussing the findings obtained and their impact in terms of contribution to the community.

**Response :**
The majority of the explored literature are based in mountainous areas (Alps, Rocky mountains, Japanese Alps) or in steppic/arctic areas (north Canada, Alaska). In addition, most of the works focus on frozen or partially frozen soils. However, in this work the soils are not frozen. Reference to the geographic conditions are added when quoting the studies.

**Comment :**
The introduction does not provide an understanding of the state of the art, or the gaps detected, which motivate this research. Although it is mentioned that it is essential to improve our understanding of the parameters that control the infiltration of melt into the soil, then papers that have addressed this problem are presented. Research questions are not research questions but,

to my understanding, define the objectives of the work. First, I understood that they could also be gaps, but after finishing reading the paper, the first option makes more sense to me. I suggest you rewrite this paragraph to clarify your questions and, also, to emphasize which are the gaps you are trying to fill with this work. In addition, the defined objective (L33) is also not addressed in the development of the work. Are the results obtained (and the analyses carried out) sufficient to conclude in this respect?

**Response :**
See above for the restructured Introduction and Result section.
In particular, the following sentence is added in the Introduction : '*In particular, additional descriptions of the partition between surface runoff and vertical infiltration of snowmelt rate into soil layers, the depth reached by vertical percolation and the depth of preferential circulations for lateral transmission along the slopes, and the response times of both the soil moisture and the river discharge, are needed.*';

**Comment :**
Modeling approaches are mentioned, and their uncertainties are discussed given the limitations in accurately representing snowmelt infiltration (L41:L42). Why not use a model in this case to evaluate the added value of the experimental setup and how it can change the perception of the processes from the comparison of the obtained results?

**Response :**
Indeed, applying a model would be a great follow-up to this work. It is cited in the conclusion as possible future work. However, this instrumental approach represents substantial work and allows results to be obtained by itself.

**Comment :**
It is emphasized that the experimental approach is novel. What makes it novel? What are the traditional instrumentation approaches? What differences are obtained in terms of process understanding? What is the added value of the proposed new instrumentation approach (cost/benefit)? It is recommended to include a benchmarking case where the proposed experimental setup is not available.

**Response :**
The novelty of this work lies in the following points:
- An important instrumental device is deployed for both snow properties/processes and soil properties/processes. Most of the study is either on snow or on the ground. No other study uses the SMA tool and at the same time offers such precise soil measurements as in this work for Alpine regions.
- 3 instrumental methods are tested and compared for measuring snowmelt. This methodological aspect fuels research on the estimation of snowmelt. Furthermore, even if they are not really new, these three tools remain relatively little applied.
- For the understanding of the processes, this work provides values on both the infiltration capacity of the soil and on the melting rates, allowing a better understanding of the partition between infiltration and surface runoff. It also provides values for typical infiltration depth of melting rate, as well as values for typical response time of soil layers and rate to melt uniformly. These results address the knowledge gap in understanding snowmelt infiltration processes on developed alpine soils highlighted in the introduction.
These 3 points are summarized at the end of the Conclusion.

**Comment :**
In L89 you mention the high uncertainty in precipitation records. Is it quantified in any way? I think a good idea (at least to check) would be to relate precipitation amounts under a threshold (e.g., 0°C) with respect to positive SWE differences (i.e. SWE(t) - SWE(t-1) > 0). With this, a relationship could be established to proxy for these biases.

**Response :**

Precipitation uncertainties were not quantified as this is not the main focus of this work. Significant work on precipitation in this same watershed has already been carried out (Benoit et al., 2018). Errors in precipitation estimation mainly come from the fact that the rain gauge is not heated. Therefore, solid precipitation is strongly underestimated. This is a limitation of the work presented in the Discussion section. However, the majority of instrumental works in the mountains come up against this same problem.

**Comment :**

Is it enough to consider LWC at the bottom of the snowpack as a proxy for snowmelt rate? What does this assumption imply? Are any conditions required for the cold content (I would expect something like CC=0)? Shouldn't we talk about "potential snowmelt"? Is it appropriate to assume that the LWC would immediately leave the bottom snow layer within the evaluated time step?

**Response :**

The hypothesis retained is that the liquid content of the lower layer of snow ($\Lambda$) is destined to infiltrate towards the ground by gravity and will not undergo additional transformation. Although a slight delay may be induced by gravitational transport through the lower layer, $\Lambda$ is used as a representation of the snowmelt flux. This hypothesis is the basic of the SMA instrument. In this work, the comparison between $\Lambda$ and the total snow water equivalent of the CRS is presented. This makes it possible to trace the limits of the initial hypothesis. These limitations are discussed in the first paragraph of the Discussion section.

**Comment :**

For the implementation of the SnowTree, is there any influence of the land cover on the measurements? Some of them are in forest areas.

**Response :**

Both SnowTrees are located in open areas, not in forest.

**Comment :**

Do the empirical thresholds defined in Eq. 1 have any theoretical basis? Are they calibrated? How sensitive are the results to different thresholds? What would be a plausible range for them? Could the same thresholds be applied in other mountain regions?

**Response :**

The limit between sensors covered by snow or not covered by snow is estimated to correspond to a 24h-standard deviation between 0.5 and 5, i.e. between the blue-ish color and the green-ish color on figure 2. 45 values have been tested for each of the two thresholds(from 0.5 to 5 with step of 0.1). The optimal value for each threshold is chosen graphically to better match the depth observed on the visual scale. Attempts of basing this optimization step on RMSE values were also performed. However, this optimization criteria did not lead to an optimal solution. Graphical selection of the optimal values is then used here. A more rubust assement of these thresholds values would be needed to ensure the validity of these thresholds in other mountain regions.

**Comment :**

When applying the cosmic ray sensor approach, how are the differences between the properties of the snowpack layers interpolated when estimating the mass? Is a single representative value obtained? Is it possible to distinguish fresh and old snow from neutron attenuation? I suggest to discuss this further and comparing it directly with the other approaches presented, otherwise it is not possible to link them.

**Response :**

The heterogeneity of the snowpack is not considered in the CRS measurement. The average neutron penetrage around the sensor is considered as a representative value. No difference is made between the different types of snow. This justify the multi-instrumental estimation of the melt

rate, either without considering the snowpack heterogeneity (CRS approach and SnowTree approach), or by focusing processes in the bottom layer.

**Comment :**
It is not clear how the selection of soil sampling points is oriented. How representative are these points with respect to the catchment? One option would be to include something like, for example, SoilGrids to have a proxy.

**Response :**
SoilGrids give a rather uniform description of the soils for he entiere catchment. The locations of the soil sampling are chosen in open grassland area where the soils where particularly deep and developed, that is to say, where the storage in the soil layer are suspected to be more important than in very shallow soils. The forest area is not represented here. Indeed, mainly other considerations would have been needed for running the study in forest and this work concentrates on alpine grassland soils. Yet, a variety of grassland have been sampled : pastured/non pastured; on flat/steep slopes, from glacial deposit/from fluvial deposit,… Table 1 synthetizes the environement of each soil sample ploint.

**Comment :**
The processes identified in L211:L215 are in line with what one would not expect and do not present a finding. Such behavior must be related to the cold content of the snowpack and the solid (liquid) precipitation on days with temperatures lower (higher) than 1°C.

**Response :**
This observation is used to illustrate the respective behavior os each of the two instrument, but indeed, this sentence can be removed.

**Comment :**
It is suggested to quantitatively support the statements made. L219 mentions the possible estimation of more precise results for melting rates, but what is the reference? It is recommended to analyze how negative SWE changes (i.e., SWE(t) - SWE(t-1) < 0) relate to the infiltration estimates made.

**Response :**
$\Lambda$ is used in this text and in the paper for the liquid water content of the bottom layer measured by the SMA.
The changes in SWE and changes in $\Lambda$ over 12 hours are represented on Figure 3. The 12h time step is represented on the figure to enhance the graphical representation, however the analysis of the two variables is also performed at the hourly time step. The SWE and $\Lambda$ do not have the same behavior at the hourly time step. In particular, the SWE reacts to the changes in temperature, whereas $\Lambda$ remains stable. This corresponds to process of transformation of snow, with a compaction of the snow pack not associated with melt. In addition, the variation of $\Lambda$ are better correlated with the variation of soil moisture than the variation of SWE at the hourly time step.
This point is detailed in the first paragraph of the Result section. Limitation of the snow melt estimation, in particular in the case of liquid precipitation on snow, is detailed in the first paragraph of the Discusion section. Last, one has to keep in mind that any field measurements remain an approximation of the reality, necessarily associated with uncertainties.

**Comment :**
Why is it important to include parameter estimation as part of the results? Wouldn't it be better to include it in the characterization of the study area? Or is soil parameter estimation also part of the instrumental setup?

**Response :**
Soil sampling in the field, as well as laser particle size analysis, represents a significant amount of work: 5 days in the field + four weeks of discontinuous analyzes in the laboratory. The values obtained provide additional knowledge for the description of the soils of Vallon de Nant and would be useful for future work in various disciplines. They represent important work and significant

results. However, for clarity, this paragraph may be merged with the soil sampling method paragraph in the Method section. Table 1 summarizes these results.

**Comment :**
Figure 3 is complex to understand because it combines, in the same panel, variables with different units and without including independent axes. This makes it difficult to analyze. What is the idea of this figure? What message is it trying to communicate?
**Response :**
The idea is to represent on the same graphe the SWE from the CRS, the snow depth from the SnowTree and the melt rate from the SMA, and also to link these variables with air temperature and precipitation. The key point is to describe the behavour of each instrument. This figure is simplified, the axis legends and the caption are enhanced for more clarity.

**Comment :**
In Figures 4 and 5, how are the positive and negative snowmelt rates interpreted? There is a lack of analysis of the results being presented.
**Response :**
Negative variations in $\Lambda$ are an increase in the liquid water content, either due to the percolation by gravity from the upper snow layer, or by actual melt within this bottom layer. These negative variations are systematically followed by positive variations in the following time step, giving this symmetrical appearance to the melt time serie. To simplify the analysis, only the positive variations of $\Lambda$ is represented on figure 4 and 5, i.e. deacresing of the snowmelt in the bottom layer due either to gravity drainage. Note that the positive variations of $\Lambda$ can also be due to a refreezing of the bottom layer in case of very thin snow cover. This is a limitation of the estimation of melt rate. However, since the SMA also provide the ice content of the bottom snow layer, refreezing periods can be determined. This point is added in the first paragraph of the Discussion section.

**Comment :**
In the analysis of the partitioning between infiltration and runoff associated with snowmelt, the effect of the different soil layers on this process is not discussed. Similarly, it is not possible to visualize what is indicated in the text in the figures referred to (e.g., L249).
**Response :**
Granolumetry analyses (and so wsat values) have been performed for each soil layer, however the results were globally very uniform. The simplication of considering the same granolumetry (and consequently the same wsat)  is done. The time period and the scale on Figure 5 are enhanced to better represent the process described.

**Comment :**
It is suggested to analyze how are the lags between precipitation peaks, flow, soil moisture and, additionally, infiltration estimates and negative SWE changes when such information is available. Currently, the analysis performed is brief and mainly qualitative (e.g., L251:L253).
**Response :**
This issue of the response time of both the soil layer and the river discharge to melt events is specifically adressed in the third paragraph of the Result section. The time periods presented on Figure 5 are modify to better represent these response times.

**Minor revisions:**

    **Comment :**
    Modify the title of the paper as it does not reflect what is being presented.
    **Response :**
    The new tittle is proposed for the paper :
    *Multi-instrumental monitoring of snowmelt infiltration in the Nant Valley, Swiss Alps*

**Comment :**
Include a table with catchment descriptors. Parameters such as area, slope, elevation range, and others are needed to interpret some results (e.g., what is the contributing rainfall area during precipitation events? -> changes in the catchment outflow hydrograph).
**Response :**
These descriptors are added in the Study area paragraph.

**Comment :**
In Table 1, include in brackets the range of variation of the estimated soil parameters. This will give an idea of the heterogeneity of each point.
**Response :**
Range of uncertainties are added in Table1.

**Comment :**
In Figure 1, is it accurate to mention geomorphological units instead of "Landforms" units?
**Response :**
This data comes from the geomorphological map of the catchment.

**Comment :**
In Figure 1 include the point where the cosmic ray sensor was installed.
**Response :**
At the Auberge station. It is added in the Figure.

**Comment :**
For clarity and readability of the document, it may be useful to combine the granulometric analysis and infiltration tests with the Soil sampling and analysis section.
**Response :**
The paragraphes in the Method section for soil analysis are reoganized.

**Comment :**
In Figure 2, considering that the y-axis depends on the scale of the SnowTree, wouldn't it be more accurate to refer to snow height?
**Response :**
'Depth' on the y-axis is replaced by 'Height'

**Comment :**
Adjust size of labels in figures. For example, in Figure 5 the labels are difficult to read or are not complete (cut off by label dimensions).
**Response :**
The appearance of all the Figure, including label and axis are revised.

**Comment :**
Correct labels and maintain consistency in abbreviations. For example, in Figure 4 Soil Moisture (HU) is mentioned in the caption, but in the panels, it is individualized differently (W?).
**Response :**
Captions are revised for all the figures. All the occurrence of HU are replace by W_r.

**Comment :**
Describe the caption in the order in which the panels are presented. For example, in Figure 4 you start with Soil Moisture, then Precipitation, Temperature, among others, and the panels are ordered as precipitation, streamflow, soil moisture.
**Response :**
The order of the variables in the caption are revised.

**Comment :**
Verify that the figures show what is indicated in the panels and captions. For example, Figure 6 shows SCA(%) and snow depth (cm) but the gray lines associated, as I understand, with SCA(%) have values above 100%.
**Response :**
Values of SCA greater than 100 % is due to the spatial resolution of MODIS.

**Comment :**
Use "snow melt" or "snowmelt", but do not combine both terms.
**Response :**
 'snowmelt' is used in all the paper.

**Authors' Response to Reviewer Comments N°3**

We thank Reviewer 3 for their thoughtful and detailed comments, which have significantly contributed to improving the clarity and scientific rigor of our manuscript. Below, we address each point raised in the review and outline the changes made to the manuscript accordingly.

**General Recommendations**

We acknowledge the need for a more coherent structure throughout the manuscript. The structure of the Introduction, as well as the structure of the Results and Discussion sections and the Conclusion are revised.

The **Introduction** has been revised to clearly present the three research questions and their significance. The structure of the introduction is now :

1. State of the art for soil processes understanding in mountain.
2. State of the art for snowmelt assessment : modeling method and the different instrumental method. The advantages and drawbacks of each method is presented.
3. Lack of knowledge in understanding in processes of snowmelt infiltration over alpine developed soils. Specifically, the following sentences are added *: 'The current understanding of the physical processes remains then insufficient to explain the role of the unsaturated zone for the infiltration of snowmelt in mountains. In particular, additional descriptions of the partition between surface runoff and vertical infiltration of snowmelt rate into soil layers, the depth reached by vertical percolation and the depth of preferential circulations for lateral transmission along the slopes, and the response times of both the soil moisture and the river discharge, are needed.';*
4. Aim of the paper. The following sentence is added : *'This work aims to address the two issues of i) providing an accurate method for the estimation of snowmelt rate, and ii) accurately detailing the physical processes involved in the unsaturated zone during the infiltration of snowmelt at a study point.'*
5. Overall methods : instrumental approach for snow and soil processes, 3 years data set, …

The **Results** section is now organized into three subsections corresponding to the research questions:

1. **Multi-instrumental estimation of snowmelt rate.**
   The results of each of the 3 snow instrument (SnowTree, CRS, SMA) are presented and compared.

2. **Partitioning of infiltration and runoff during snowmelt.**
   The intensities of melt rates are compared with the infiltration capacity of soils to better quantify the partition between surface runoff and infiltration of the melt flux, in winter and in spring season.

3. **Vertical and lateral transfers of meltwater and its impact on streamflow.**
   The vertical percolation of the melt flux into soil layers and its response to melt events are described. The typical response times of the soil layers at the three instrumented sites in the catchment to melt event are described. The response time of river discharge is also presented.

- The **Discussion** section revisits each research question, providing detailed interpretations of the results and connecting them to broader implications. The

paragraphs that were previously mixed into the Results section (e.g., lines 216–223) have been relocated and expanded within the Discussion.

The Discussion section is now organized as follow :

**1. Uncertainties in snowmelt estimation**
The limit of each snow instrument is presented, and their complementarity for snowmelt assement is presented. In particular, the impacts of rain-on-snow phenomenon is investigated.

**2. Inter-annual variability of the snow contribution**
The variability between the 3 recorded years is described. Divergences and common behaviors within the recorded seasons are drawn. A open discussion about the potential impat of climate change on the presented results if raised.

**3. Link with undeground water storage**
The issue of the interaction with deep storage is detailed. A discussion about the possible futur works about groundwater investigation in the catchment is open.

- The **Conclusions** is restructured to explicitly summarize the answers to the research questions and highlight the contributions of the study. The Conclusion section is now organized as follow :
    1. Reminder of the objectives of the paper and the three main points of the results
    2. A quick summary of the main results, organized along the same structure than the Result section (3 bullet points)
    3. The possible application of this work as validation/calibration data for futur modeling works is highlighted.

**Additional Revisions**
We have carefully reviewed the annotated PDF provided by the reviewer and incorporated the suggested edits where appropriate.

**Comment :**
- L2:You emphasize the importance of parameters controlling snowmelt infiltration here, as well as in the introduction. What are the specific parameters you investigated? It feels a bit unclear to me what the word "parameters" refers to.
**Response :**
The study focus on the following parameters/variables are : i) the partition between surface runoff and vertical infiltration of snowmelt rate into soil layers, ii) the depth reached by vertical percolation and the depth of preferential circulations for lateral transmission along the slopes, and iii) the response times of both the soil moisture and the river discharge, are needed.
This is clarified both in the abstract and in the introduction

**Comment :**
- L9 :This phrasing is a bit unclear. What does point measurement mean?
**Response :**
'At the point measurement' is replaced by 'At the location of the instrumented site'

**Comment :**

- L10-11 :This concluding sentence does not logically draw from the findings presented in the earlier part of the abstract. I recommend modifying this sentence so that it more clearly draws a conclusion from the results as presented, or add another sentence to provide greater context. Why do fast response times of shallow soil moisture to snowmelt emphasize that mountain areas will be vulnerable to dry periods in the future?

**Response :**

This sentence in the abstract is replaced by : 'These findings highlight how the fast response times of shallow soil moisture to snowmelt may limit the capacity of mountain soils to retain water, potentially increasing their vulnerability to dry periods in the future. '

**Comment :**

-L19 :again, please clarify what parameters are important/what this refers to.

**Response :**

See the corresponding response for the abstract.

**Comment :**

- L55- 60 : One potential limitation with this approach is that you are tracking celerities, not velocities of water. It would be good to explore the impact of celerities vs. velocities.

**Response :**

The origin of water (does soil water content and discharge answer to snowmelt with the actual melted volume or with former/deeper storage?) is indeed an important question. The difference between young water (from the year) and deeper storage is well described by isotopic studies. In this work, the soil layers are shown no react quickly to melt event. Consequently, no long term storage are observed in the superficial soil (above 30 cm). Water measured in soil are then necessarily originated from the seasonal snow. Regarding the answer of discharge to melt event, the respective contributions of surface runoff, soil water and groundwater are indeed not investigated in this work. This question is raised in the third paragraph of the Discussion section ( Link with undeground water storage). The two given reference are added there.

**Comment :**

- L100  : Are there other studies that have employed this device besides proprietary studies from the manufacturer?

**Response :**

The SMA is usually used in combination with a more complete instrumental setup for snowpack analysis, the Snow Pack Analyzer (SPA). Various studies use the SPA. For exemple :

-Schattan, P., Baroni, G., Oswald, S. E., Schöber, J., Fey, C., Kormann, C., ... & Achleitner, S. (2017). Continuous monitoring of snowpack dynamics in alpine terrain by aboveground neutron sensing. *Water Resources Research*, *53*(5), 3615-3634.

-Fromm, R., Baumgärtner, S., Leitinger, G., Tasser, E., & Höller, P. (2018). Determining the drivers for snow gliding. *Natural Hazards and Earth System Sciences*, *18*(7), 1891-1903.

-Egli, L., Jonas, T., & Meister, R. (2009). Comparison of different automatic methods for estimating snow water equivalent. *Cold Regions Science and Technology*, *57*(2-3), 107-115.

These references are added in the text.

**Comment :**

-L103  : Is this an established assumption? Or can you provide some rationale for the assumption?

**Response :**

This assumption is the base of the SMA instrument. The assumption made is that the liquid content

of the bottom snow layer is bound to percolate toward the soil by gravity and will not undergo further transformation. Even though a slight delay may be induced by gravity transport through the bottom layer, the liquid content of the bottom snow layer is used as a representation of the snowmelt flux. In this work, the comparizon between the liquid content of the bottom snow from the SMA and the total Snow Water Equivalent by the CRS is presente. This helps to draw the limits of the initial assumption. This limits are discused in the first paragraph of the Discussion section.

**Comment :**

-L165: since you use the names of the sites above, you should be consistent and list the names of the sites here rather than saying generically "three points in the catchment"
**Response :**
 'three points in the catchment' is replaced by 'Auberge, Chalet and La Chaux sites'.

**Comment :**
-L194-205 : These results discuss how well the snow tree measurements work, but there's no actual interpretation of the data itself. Were there differences between sites? What were the average snow depths and how did they vary with time?
**Response :**
The results of the SnowTree at the Auberge and the Chalet site are first presented. The obtained snow depths are then used in Figure 3, for comparison with the SMA and CRS results.  A description of the dynamics of the snow depth is added in the text.The main difference between the SnowTree at the Auberge and the Chalet sites is the influence of solar heating (the Auberge site is more shaddowed than the Chalet site). This is specified in the text.

**Comment :**
-Figure 3 : If you're going to describe every variable, you need to start at the top panel and work your way to the bottom, describing everything sequentially. You need to adjust the order of the figure legend so that it matches the order of the panels and the caption. Super confusing as written.
**Response :**
The appearance of the figures is completely revised. The order of the panel is changed.

**Comment :**
-L.237 : more neutral/scientific word choice needed. intermittent maybe?
**Response :**
'chaotic' is replaced by ' intermittent'

**Comment :**
-Figure 4 : Again, the order of the caption is super confusing and does not match the figure. You list soil moisture as HU, but there is no y-axis labeled HU? Is it supposed to be the axis labeled W? I'd recommend SM as an abbreviation for soil moisture, or VWC for volumetric water content. Fix the order of the figure to match the caption. Make the colored lines in the legend thicker, I can't see them.
**Response :**
The appearance of the figures is completely revised. The order of the panel is changed. The thickness of the legend if modified. Since no multi-letter variable name is recommended by the edition, W and W_r are chosen for volumetric water content and relative volumetric water content. The occurrence of HU are corrected.

**Comment :**
-L270 : What site is this for?
**Response :**
For the Auberge site. This is precized in the text.

---

## Author Response (AR2)

**Authors' Response to Reviewer N°2's minor revisions**

Minor revisions have been made as requested. Figure readability has been improved. Grammar and syntax have also been corrected.